# Benchmark decadal forecast skill for terrestrial water storage estimated by an elasticity framework

Enda Zhu [1,2], Xing Yuan [1,3] & Andrew W. Wood[4]

A reliable decadal prediction of terrestrial water storage (TWS) is critical for a sustainable management of freshwater resources and infrastructures. However, the dependence of TWS forecast skill on the accuracy of initial hydrological conditions and decadal climate forecasts is not clear, and the baseline skill remains unknown. Here we use decadal climate hindcasts and perform hydrological ensemble simulations to estimate a benchmark decadal forecast skill for TWS over global major river basins with an elasticity framework that considers varying skill of initial conditions and climate forecasts. The initial condition skill elasticity is higher than climate forecast skill elasticity over many river basins at 1–4 years lead, suggesting the dominance of initial conditions at short leads. However, our benchmark skill for TWS is significantly higher than initial conditions-based forecast skill over 25 and 31% basins for the leads of 1–4 and 3–6 years, and incorporating climate prediction can significantly increase TWS prediction skill over half of the river basins at long leads, especially over mid- and high-latitudes. Our findings imply the possibility of improving decadal TWS forecasts by using dynamical climate prediction information, and the necessity of using the new benchmark skill for verifying the success of decadal hydrological forecasts.

[1] Key Laboratory of Regional Climate-Environment for Temperate East Asia (RCE-TEA), Institute of Atmospheric Physics, Chinese Academy of Sciences, 100029 Beijing, China. [2] College of Earth and Planetary Science, University of Chinese Academy of Sciences, 100049 Beijing, China. [3] School of Hydrology and Water Resources, Nanjing University of Information Science and Technology, Nanjing, 210044 Jiangsu, China. [4] Research Applications Laboratory, NCAR, Boulder, CO 80301, USA. Correspondence and requests for materials should be addressed to X.Y. (email: xyuan@nuist.edu.cn)

errestrial water storage (TWS), including snow and ice, surface water stored in reservoirs, lakes and rivers, and underground water in vadose zones and aquifers, is critical for the global hydrological cycle and freshwater resources[1–3]. Its variations not only influence weather and climate significantly through a series of complex processes and feedbacks[4], but also change sea level and affect Earth's rotation[5,6]. Therefore, predicting TWS change a few years in advance can provide invaluable information for additional sources of climate predictability, and for a sustainable management of water resources and infrastructures[7,8]. However, decadal prediction of TWS is at an exploratory stage[9] due to limited knowledge of hydrological predictability and forecast skill at interannual to decadal scales, and a benchmark skill is needed to guide the applications of decadal hydrological predictions.

Hydrological predictability mainly comes from two sources: the memory in initial hydrological conditions (IHCs) and the predictability of meteorological forcings[10–14]. IHCs can be significant sources of hydrological predictability even at interannual to decadal scales. Based on the information in IHCs alone, TWS over one third of the world's land areas could be skillfully predicted 2–5 years ahead, especially over semiarid regions such as northern China, southern Africa, and the Middle East, where the hydrological variability is not negligible and the hydrological anomalies can persist for a long time[9]. Besides IHCs, the success of decadal climate forecasts (DCFs) determines the skill of TWS prediction at long leads. DCFs rely on the understanding of the low frequency of climate variability (e.g., Pacific Decadal Oscillation and Atlantic Multidecadal Oscillation), natural forcings such as volcanic activities, and human-induced climate change through the emissions of anthropogenic aerosols and greenhouse gases. These internal and external factors affect temperature and precipitation through complicated ocean-atmosphere teleconnections and cloud-aerosol-radiation interactions. Decadal prediction has been regarded as one of the seven grand challenges by the World Climate Research Program, with gradual improvement during past 20 years[15,16]. Recently, a number of international projects, including the Decadal Climate Prediction Project[17], which will contribute to the sixth Coupled Model Intercomparison Project[18], have been launched to investigate decadal climate predictability and variability, and to provide experimental quasi-real-time decadal prediction. Given the progress in utilizing the memory in IHCs and the information in DCFs, the era for decadal hydrological prediction is expected in the near future.

To distinguish the effects of IHCs and meteorological forcings (e.g., DCFs) on the hydrological forecasts, simulations based on the ensemble streamflow prediction (ESP) method that uses accurate (deterministic) IHCs but climatological ensemble meteorological forcings, were compared with simulations based on the reverse ESP (rev-ESP) method that uses accurate (deterministic) meteorological forcings but climatological ensemble IHCs[19–21]. Comparing the performances of ESP and rev-ESP simulations enables estimation of the influence of IHCs relative to that of boundary conditions—e.g., DCFs[9]. In past demonstrations and applications, this framework only considered perfect-model experiments—that is, treating IHCs (i.e., ESP) or meteorological forcings (i.e., rev-ESP) as observations for the purpose of determining relative errors. However, in reality, there are variable uncertainties in IHCs and climate forecasts depending on the characteristics of the hydroclimate system. Therefore, the concept of forecast skill elasticity framework has been recently proposed to investigate the streamflow predictability at seasonal time scale[22,23], where the accuracy of IHCs and DCFs can vary from 1 (perfect) to 0 (climatology). The forecast skill elasticities[22] (i.e., Eqs. 3 and 4 in Methods) represent the gradients in hydrological

forecast skill (e.g., streamflow forecast skill) relative to gradients in accuracy for predictors (e.g., IHCs or meteorological forcings), hence a larger elasticity of a predictability source means a larger contribution to the improvement of the hydrological prediction given an improved skill in the predictability source.

However, whether such an elasticity framework is applicable for quantifying potential benefit of improving IHCs and DCFs for decadal TWS prediction over global major river basins needs further investigation. In addition, there is an opportunity to obtain a benchmark forecast skill by incorporating the-state-of-the-art information of DCFs from the fifth Coupled Model Intercomparison Project (CMIP5) decadal hindcasts[24]. Prior studies and operational applications have employed post-ESP or statistical-dynamical (hybrid and hierarchical) approaches to incorporate climate information into the seasonal hydrological forecasting[25,26]. Here, we propose a method for estimating a benchmark decadal hydrological prediction skill based on a climate-hydrology approach[11] (e.g., physical hydrological model predictions driven by CMIP5 decadal climate predictions), and further use the elasticity framework to quantify predictability gradients. The benchmark skill can provide a new norm or starting point to assess whether different strategies for future upgrades in the climate-hydrology approach[11] are beneficial for decadal hydrological prediction.

## Results

**TWS forecast skill elasticity over global river basins.** To analyze TWS prediction skill with different levels of accuracy in IHCs and DCFs, we applied the ESP and rev-ESP methods by resampling[23] the ensemble simulations (see Methods and Supplementary Figs. 1–3 for details). The ESP and rev-ESP experiments were carried out by performing Community Land Model version 4.5 (CLM4.5)[27] simulations. In these experiments, the DCFs include atmospheric temperature, humidity, wind and pressure near the surface, radiation, and precipitation, while IHCs represent the initial states of TWS, soil moisture and soil temperature, etc. Along the $x$-axis in each panel in Fig. 1, the Nash-Sutcliffe Efficiency (NSE; see Methods for details), a metric of skill for IHCs, increases from the left to the right with the ending point, ESP (representing perfect initial conditions together with climatological DCFs). Along the $y$-axis, the DCFs NSE increases from bottom to top with the ending point, rev-ESP (representing perfect meteorological forcings (DCFs) together with climatological IHCs). The upper right corner for each panel shows a perfect TWS forecast (both IHCs and DCFs are perfect) with a NSE value of one, while the lower left corners (climo, abbreviation for climatology) show forecast skill based on climatological information only, resulting in NSE values of zero or less (Fig. 1a).

As the forecast lead time increases from 1–4 years to 7–10 years (here, 1–4 years lead prediction represents 4-year average prediction with 0-year lead, 2–5 years lead prediction represents that with 1-year lead, and so on), the increasingly horizontal contours show that the NSE skill becomes less sensitive to the IHCs uncertainty and is more dominated by DCFs, as the prediction becomes more of a boundary value than an initial value challenge (Fig. 1). For instance, when the DCFs NSE is about 0.5 at 5–8 years lead, there is no obvious improvement for TWS prediction even if the IHCs NSE increases from 0 to 1 for the Amazon basin (Fig. 1c). However, at 1–4 years lead (Fig. 1a), increasing IHCs NSE results in significant improvement of TWS prediction, given the same DCFs NSE (e.g., 0.5). The insensitivity of TWS skill to the increasing skill in IHCs is more obvious for the Amazon basin in the humid area (Fig. 1a–d) than the other two basins (Fig. 1e–l), while the TWS skill sensitivity to increasing IHCs skill for the Yenisei basin in the semiarid area (Fig. 1i–l, see

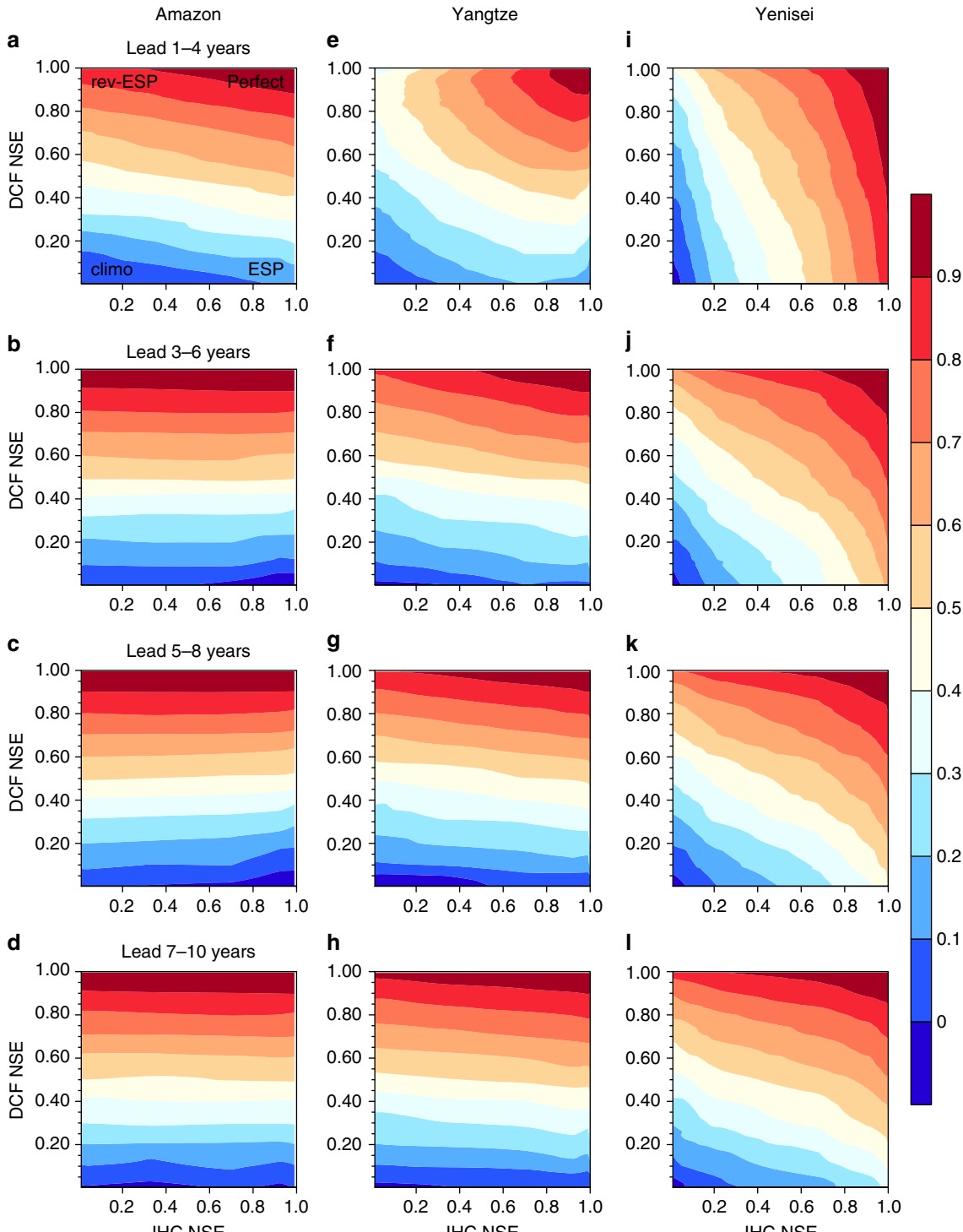

**Fig. 1** The decadal hindcast skill for terrestrial water storage with varying skill of initial conditions and climate forecasts. The results are for 4-year mean basin-averaged terrestrial water storage over three selected basins (**a–d** Amazon; **e–h** Yangtze; **i–l** Yenisei) at different lead times. x- and y-axes represent skill for initial hydrological conditions (IHCs) and decadal climate forecasts (DCFs). The skill used here is the Nash-Sutcliffe efficiency (NSE), with a value of one representing a perfect forecast, and a value less than zero representing a forecast poorer than a climatological forecast

Supplementary Fig. 1 for the locations) is the largest. This suggests that increasing IHCs skill has significant potential benefit to improve decadal prediction of TWS over the Yenisei basin, while the control for IHCs drops quickly in the first few years for the Amazon basin. The results for the Yangtze basin, a semi-humid river basin, fall between the humid and semiarid basins

(Fig. 1). Similar characteristics for skill variations are also found by using correlation as the metric (Supplementary Fig. 4).

Given a range of skill in IHCs or DCFs, the corresponding TWS forecast skill elasticity can be estimated (see Methods for details) for each river basin at each forecast lead. The elasticity can quantity the contribution of improved skill in predictability

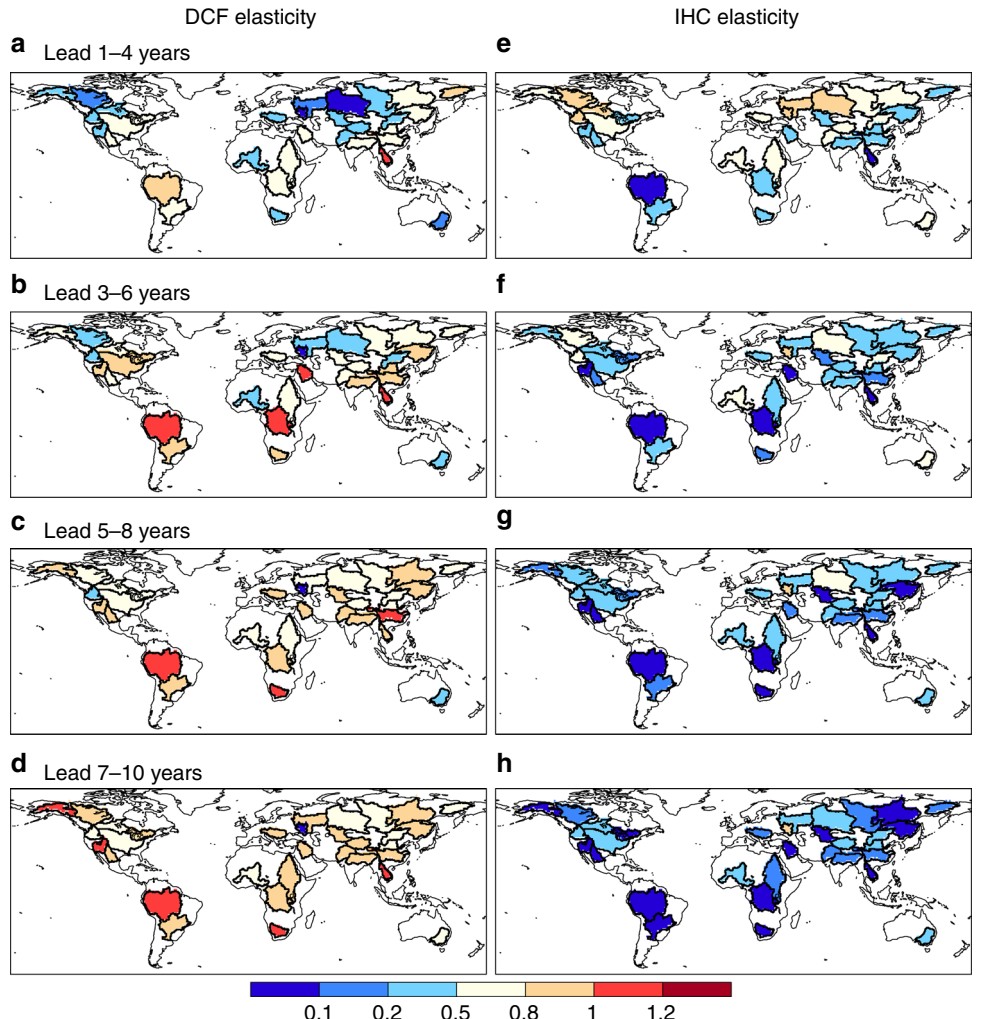

**Fig. 2** Skill elasticities for climate forecasts and initial conditions. The skill elasticities were calculated by using Eqs. 3 and 4 in the Methods for 4-year mean terrestrial water storage (TWS) hindcasts at different lead times over 32 major river basins, both for **a–d** decadal climate forecasts (DCFs) and **e–h** initial hydrological conditions (IHCs). For example, a value of 0.8 represents that 100% improvement in the accuracy of DCFs or IHCs (e.g., Nash-Sutcliffe efficiency (NSE) increases from 0.2 to 0.4) would result in 80% increase in TWS forecast skill (e.g., NSE increases from 0.3 to 0.54). Maps were created by using the NCAR Command Language (Version 6.3.0) [Software]. (2016). Boulder, Colorado: UCAR/NCAR/CISL/TDD. https://doi.org/10.5065/D6WD3XH5

sources (e.g., IHCs or DCFs) to the improvement of TWS prediction. Figure 2 shows skill elasticity over 32 global major river basins with forecast lead times from 1–4 years to 7–10 years. As the forecast lead time increases, the IHCs skill elasticity decreases but the DCFs skill elasticity increases. At shorter leads (e.g., 1–4 years), the IHCs' skill elasticity is higher than DCFs' skill elasticity over half of the river basins, especially in high-latitude and arid and semiarid regions (e.g., Yenisei, Ob, Mackenzie, Niger, and Nelson) (Fig. 2a, e). This suggests that increasing the accuracy in IHCs can bring more benefit to TWS decadal prediction in these regions. On the contrary, DCFs dominate the predictability in humid regions. For instance, the DCFs skill elasticity is consistently higher than the IHCs skill elasticity for the Amazon and Yangtze basins even for the first 4 years forecasts, but the IHCs skill elasticity can be dominant up to 5 years over the Yenisei basin (Supplementary Fig. 5). Averaged over the major river basins, the DCFs NSE (correlation) elasticities increase from 0.48 (0.36) to 0.81 (0.73) as lead time increases from 1–4 years to 7–10 years, while the IHCs NSE (correlation) elasticities decrease from 0.55 (0.36) to 0.19 (0.13) (Fig. 2 and Supplementary Fig. 6). DCFs elasticities are larger

than one over 15% (22% for correlation) of the basins at 7–10 years lead, suggesting great benefits for advancing TWS long-lead prediction by improving decadal climate prediction.

**Benchmark decadal forecast skill for TWS.** With the skill elasticity framework, we can estimate the decadal forecast skill for TWS given a DCFs skill or IHCs skill (Fig. 1). For example, when the NSE skill for meteorological forcings is equal to 0.2 and IHCs are perfect, the TWS prediction skill is about 0.25 for the Amazon basin at lead 1–4 years (Fig. 1a).

To estimate the actual decadal forecast skill for meteorological forcings, ten CMIP5 models were selected (see Methods for details). The NSE values for the multi-model ensemble mean precipitation averaged over global major river basins at different lead times are shown in Fig. 3. For the 1–4 years lead, there are skillful predictions (NSE > 0 and correlation larger than 0.5) over the Amazon basin in South America, the Shatt el Arab basin in Middle East, the Ob and Syr-Darya basins in central Asia, and the Lena basin in Far East (Fig. 3a and Supplementary Fig. 7a). The predictions are also skillful over certain regions even at long leads, such as the North American and high-latitude river basins at the

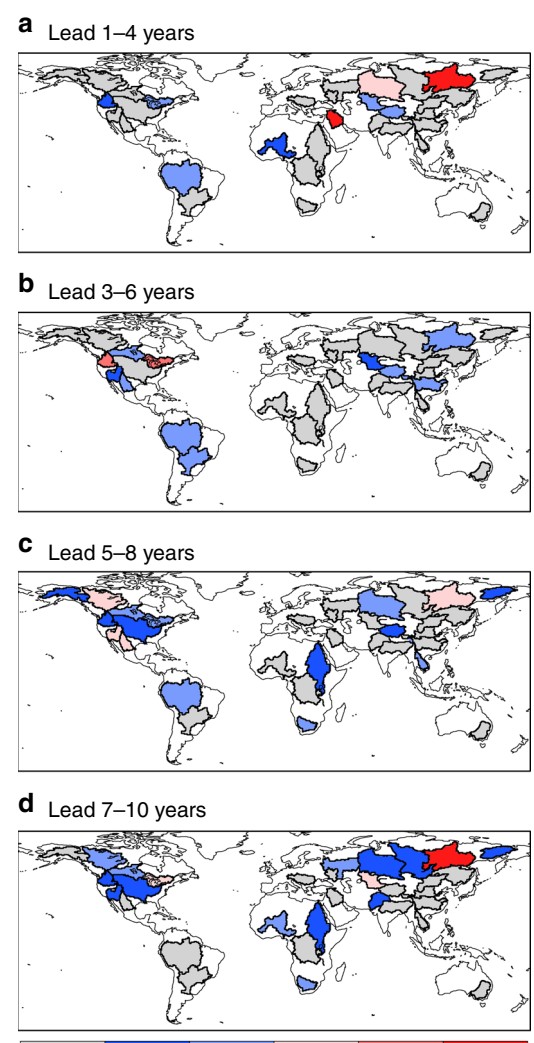

**Fig. 3** Decadal hindcast skill for basin-averaged precipitation. Nash-Sutcliffe efficiency (NSE) values were calculated for CMIP5 multi-model-predicted 4-year mean precipitation averaged over global major river basins at different lead times (**a–d**). The anomaly of CMIP5 model-predicted precipitation to CMIP5 climatology was used to calculate NSE, to circumvent the bias in the interpretation of results. Maps were created by using the NCAR Command Language (Version 6.3.0) [Software]. (2016). Boulder, Colorado: UCAR/NCAR/CISL/TDD. https://doi.org/10.5065/D6WD3XH5

7–10 years lead (Fig. 3b–d). The prediction skill does not necessarily decrease over leads, which might be caused by the perturbations from external forcings[24]. However, these external forcings, such as volcanic eruptions, are unpredictable before their occurrence[28], and their effects on the skill variation may be amplified given limited decadal hindcast samples.

After obtaining the actual DCFs skill based on the CMIP5 hindcasts analysis above, we can estimate the benchmark skill, which considers skill both in IHCs and DCFs comprehensively, for the decadal prediction of TWS by using the elasticity framework. At 1–4 years lead, the benchmark skill is beyond 0.7 over 25% basins (8 in 32 basins) (Fig. 4a), which are located in high-latitude (e.g., Volga, Yenisei and Lena) or semiarid regions (e.g., Niger and Nile) where memory of IHCs is important for hydrological predictability. As forecast lead time increases, the benchmark skill averaged over the 32 major river basins decreases from 0.51 (0.67 for correlation) at 1–4 years lead (Fig. 4a and

Supplementary Fig. 8a) to 0.25 (0.42), 0.19 (0.30), and 0.11 (0.17) at leads of 3–6 years, 5–8 years, and 7–10 years, respectively (Fig. 4b–d and Supplementary Figs. 8b–d). Compared with ESP forecast skill, which has been used for the benchmark in most hydrological applications (Fig. 4e–h and Supplementary Figs. 8e–h), the new benchmark skill is significantly ($p < 0.05$) higher over 25% (43% for correlation) river basins after incorporating CMIP5 decadal prediction information at 1–4 years lead. The increases are more obvious at longer lead time, with significant NSE (correlation) increases over 31% (56%), 44% (66%), and 47% (59%) basins at leads of 3–6, 5–8, 7–10 years.

## Discussion

Decadal TWS prediction can be influenced by both IHCs and DCFs. With the improvement in DCFs, a more skillful TWS prediction is expected. In this study, we provide a benchmark skill for TWS decadal prediction by comprehensively considering skill of both DCFs from CMIP5 decadal prediction models and perfect IHCs through an elasticity framework. Compared with the ESP that is regarded as a benchmark for most hydrological forecasting studies, our proposed benchmark significantly increases skill (NSE) over 25% and 31% basins for the 1–4 years and 3–6 years TWS prediction respectively. The average NSE for the ESP predicted TWS is close to zero beyond 5 years, but the benchmark skill (NSE) is 0.19 and 0.11 for the TWS predictions at 5–8 years and 7–10 years lead. In addition, the elasticity framework based on the coordinate transformation (see Methods for details) can be applied to analyze the major sources of hydrological decadal predictability, as well as the actual forecast skill enhancement given improvements in IHCs and/or DCFs. Our benchmark skill provides a new norm to guide the application of decadal hydrological predictions.

A reliable hydrological prediction provides a valuable reference for optimal operation and management of water resources, therefore, hydrological skill elasticity can suggest whether investing on IHCs or DCFs can bring more benefit, and which way is the most cost-effective. The major source of hydrological predictability is different over different basins at different lead times. Therefore, whether improving IHCs or DCFs is more effective depends on specific situations. For example, the elasticity analysis shows that improving IHCs skill can bring more potential value than improving DCFs skill over northern parts of Eurasia, North America and Africa at 1–4 years lead (Fig. 2).

Here, we calculate NSE instead of variational weights (fraction of climatology)[22,23] to make the elasticity framework more straightforward for assessing actual skill. Through coordinate transformation, we can obtain the TWS prediction skill in the background of CMIP5 DCFs skill. It is viewed as a new benchmark, including information not only from IHCs but also from the state-of-the-art DCFs. In addition, we can obtain the benchmark skill for decadal TWS prediction through the elasticity framework instead of downscaling CMIP5 DCFs and driving a land surface hydrological model to produce TWS prediction that is similar to most seasonal hydrological forecasting studies[11]. In other words, the benchmark skill estimated in our study can be regarded as a new reference for verifying the usefulness of any seasonal to decadal hydrological forecasts, regardless of using complicated downscaling approaches such as Bias Correction and Spatial Downscaling method[29], Bayesian method[30], or even dynamical downscaling[31]. This paper analyzed TWS decadal predictability and benchmark skill, but it can be applied to other hydrological variables (e.g., streamflow, soil moisture) from subseasonal to decadal scales.

## Methods

**Study basins**. This study analyzed the TWS predictability and prediction skill over global major river basins[32]. The selected 32 river basins are the focus of the Global

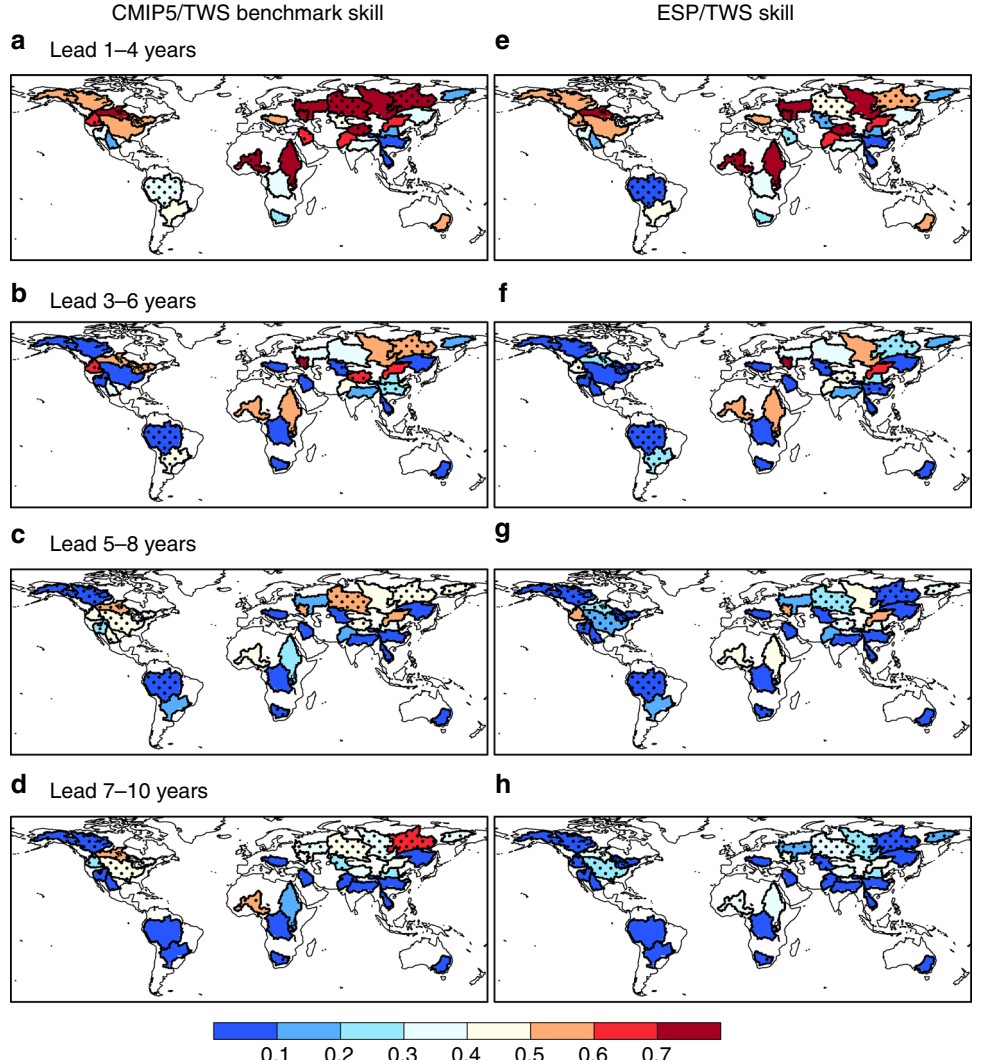

**Fig. 4** Comparison between benchmark decadal hindcast skill and initial conditions-based hindcast skill. The results are for 4-year mean terrestrial water storage (TWS) averaged over global major river basins. Here, the CMIP5/TWS benchmark hindcasts (**a**–**d**) were obtained by using the TWS skill elasticity analysis with decadal climate forecast skill specified by CMIP5 multi-model-predicted precipitation, while initial conditions-based hindcasts (**e**–**h**) were from ensemble streamflow prediction (ESP) simulations. The dotted regions represent river basins with benchmark skill that is significantly ($p < 0.05$) higher than ESP skill, estimated by using bootstrapping 1000 times. Maps were created by using the NCAR Command Language (Version 6.3.0) [Software]. (2016). Boulder, Colorado: UCAR/NCAR/CISL/TDD. https://doi.org/10.5065/D6WD3XH5

Energy and Water EXchanges (GEWEX) project, representing a broad range of climate and land cover conditions (Supplementary Fig. 1). Three basins (i.e., Amazon, Yangtze, and Yenisei) were selected to illustrate details of variable uncertainty in IHCs and DCFs, and their corresponding TWS predictability over different basins. The Amazon basin is located in the low latitude with an area of 5,854,000 km$^2$, where the annual rainfall is abundant. The Yangtze basin is located in the middle latitude, covering 1,794,000 km$^2$ where the amount of rainfall is less than the Amazon, and there are significant interannual variations for precipitation due to the East Asian Summer Monsoon activities. The Yenisei basin is mainly located in Russia, a high-latitude basin with an area of 2,579,000 km$^2$ and a continental subarctic climate. In this study, all the statistics were calculated based on basin average values.

**CMIP5 decadal climate predictions.** Precipitation outputs from the decadal hindcasts of 10 CMIP5 models, i.e., bcc-csm 1-1, MRI-CGCM3, CNRM-CM5, EC-EARTH, IPSL-CM5A, FGOALS-g2, FGOALS-s2, MIROC4h, MPI-ESM-LR, MPI-ESM-MR, were equally averaged to form a multi-model ensemble mean to estimate decadal prediction skill for precipitation[33], which was used as the DCFs skill in this work. For each CMIP5 model, the r1i1p1 realization was used. The hindcasts started every 5 year from 1960 to 2005, and ran over 10 years periods. The century-long Climatic Research Unit-National Centers for Environmental Prediction (CRUNCEP[34]) observed meteorological forcings data was used to drive the land surface model, and was used as a reference to assess the performance of precipitation predictions from CMIP5 models. The dataset is a widely used gauge-

based product, and its monthly precipitation values are the same as the CRU data. As compared with other precipitation products based on gauge-observation or satellite-estimates, including CPC, GPCC, GPCP, and PREC, regional differences in precipitation magnitude do exist between CRU and other products (Supplementary Fig. 9). However, to be consistent with ESP and rev-ESP experiments for elasticity and benchmark skill estimation which are based on CRUNCEP-driven CLM4.5 simulations, we kept using CRUNCEP as a reference for validating CMIP5 decadal prediction of precipitation. The time series of decadal precipitation forecasts along with the time series of observed precipitation for each river basin are shown in Supplementary Fig. 10.

**Prediction skill.** We use the Nash-Sutcliffe Efficiency (NSE) to assess the skill. NSE is calculated as:

$$\text{NSE}_q = 1 - \left[ \frac{\sum_{i=1}^{n} \left( x_{i,q}^{'\text{obs}} - x_{i,q}^{'\text{sim}} \right)^2}{\sum_{i=1}^{n} \left( x_{i,q}^{'\text{obs}} \right)^2} \right] \tag{1}$$

where $q$ is forecast lead (years; e.g., 1–4 years, 2–5 years), $i$ stands for the $i$th initialization year (i.e., 1961, 1966, 1971, 1976, 1981, 1986, 1991, 1996, 2001, 2006), $x_{i,q}^{'\text{obs}}$ is the anomaly of observation for the $i$th initialization year at lead $q$ years for the concerned component (e.g., precipitation or TWS), $x_{i,q}^{'\text{sim}}$ is the same as $x_{i,q}^{'\text{obs}}$ but

for the anomaly of simulations or predictions (we use anomaly to avoid the impact of prediction bias), and $n$ ( $= 10$ in this study) is the total number of initialization years in the CMIP5 decadal hindcasts. NSE spans from $-\infty$ to 1, with NSE $= 1$ means a perfect prediction. The values between 0 and 1 are generally viewed as acceptable levels of performance, but the values <0 means the climatological forecast is even better than model forecast, which indicates unacceptable performance. NSE is commonly used in the hydrological field[35].

**ESP and rev-ESP**. In this work, decadal hydrological simulations and predictions[11] were produced by CLM4.5. Terrestrial water storage (TWS), including liquid and solid soil moisture, unconfined aquifer water, canopy water, snow water, river water storage, and surface water storage for wetlands and small subgrid-scale water bodies[27], was used to illustrate the elasticity framework. Previous studies have verified that the model is able to simulate large-scale TWS dynamics reasonably[36].

A continuous CLM4.5 control simulation (CTL) was driven by CRUNCEP data during 1901–2010 to provide the IHCs and verification data, viewed as a true situation. The first 50-year simulations during 1901–1950 were used as land surface model spin-up. Similar to CMIP5 decadal climate hindcast experiments, the decadal hydrological hindcasts were designed to start every 5 years from 1951 to 2001 by using IHCs obtained from CLM4.5 CTL experiment, and were run over 10-year periods. As shown in Supplementary Fig. 2, both ESP and rev-ESP simulations were performed. The ESP-type simulations, which we regard as real decadal hydrological hindcasts based on IHCs alone, have ten ensemble members with the same (true) initial hydrological conditions from CTL experiments but different meteorological forcings from 10-year samples overlapped every 5 years during 1951–2001 (excluding that started from the target year). Rev-ESP[21] simulations have a set of initial conditions sampled from the study period (excluding the target years) but with the same (true) meteorological forcings.

**Elasticity framework**. The IHCs and DCFs for ESP and rev-ESP are either perfect or climatological. To analyze TWS predictability under different accuracy levels of IHCs and DCFs, the predictions were produced via the linear combination of the climatological and perfect IHCs/DCFs. The weights used for combinations were ($w = 0, 0.05, 0.10, 0.25, 0.50, 0.75, 0.90, 0.95, 1.0$)[22]. A weight ($w_{IHC}$ or $w_{DCF}$) equal to zero yields a perfect condition and equal to unity means the climatological knowledge. The skill of TWS prediction for each $w_{IHC}$ and $w_{DCF}$ combination can be estimated. As shown in Supplementary Fig. 3a, the blue dots are the end points of the variable weight assessment: the ESP, the perfect forecast, the rev-ESP, and the climatological forecast. By using end points blending (EPB) method[23], we combined the four end points to generate hindcasts for each $w_{IHC}$ and $w_{DCF}$ combination without running additional simulations. For each combination point, we blended the four end points based on the given weight to generate a 100-member ensemble, and we calculated the percentage of each end point (EP(%); i.e., the number of members randomly selected from each end point) using the following equation:

$$\mathrm{EP}(\%) = \left(1 - |x_{EP} - w_{IHC}|\right) \times \left(1 - |y_{EP} - w_{DCF}|\right), \qquad (2)$$

where $w_{IHC}$ and $w_{DCF}$ are the weights for IHCs and DCFs respectively, $x_{EP}$ and $y_{EP}$ are the $w_{IHC}$ and $w_{DCF}$ values at the end points for which the percentage is calculated (e.g., $x_{EP} = 1$ and $y_{EP} = 0$ are for the ESP-point). This procedure was carried out for each 10-year simulation and for each basin. Through the method mentioned above, we obtained a surface plot for TWS skill in terms of NSE, where $x$- and $y$-axis are the weights in IHCs and DCFs, respectively (Supplementary Fig. 3a). Then, we used the same method to generate 100 members for each weight value on the axis, and calculated the NSE for DCFs and IHCs to replace the specified weights. Taking 1966-01 simulation for Amazon basin as an example, for the weight = 0.25 for $y$-axis, we chose 25 sets of 10-years forcings randomly from the ten groups (without 1966–75) of meteorological forcings (1951–61, 1956–65, 1961–70, 1971–80, 1976–85, 1981–90, 1986–95, 1991–2000, 1996–2005, 2001–2010) from CRUNCEP, and 75 sets of real forcings (1966–75). Thereafter, we calculated the NSE for DCFs to replace the 0.25 on $y$-axis (Supplementary Fig. 3b). As a result, the surface plot was transformed to another coordinate, representing NSE skill for IHCs and DCFs (Supplementary Fig. 3b). As the random resampling produced random biases, we did a 9-point smoothing along the $y$-axis to generate a more uniform response surface. Finally, we calculated the gradients in TWS prediction skill relative to IHC and DCF skill from a number of blending points (shown with plus plotting symbols in Supplementary Fig. 3b), and we called them as decadal hydrological prediction skill elasticity:

$$E_{IHC} = \left\{ \begin{array}{c} \frac{NSE[TWS(0.50,0.90)] - NSE[TWS(0.90,0.90)]}{NSE[IHC(0.50)] - NSE[IHC(0.90)]} \\ + \frac{NSE[TWS(0.50,0.75)] - NSE[TWS(0.90,0.75)]}{NSE[IHC(0.50)] - NSE[IHC(0.90)]} \\ + \frac{NSE[TWS(0.50,0.50)] - NSE[TWS(0.90,0.50)]}{NSE[IHC(0.50)] - NSE[IHC(0.90)]} \end{array} \right\}/3 \qquad (3)$$

$$E_{DCF} = \left\{ \begin{array}{c} \frac{NSE[TWS(0.90,0.50)] - NSE[TWS(0.90,0.90)]}{NSE[DCF(0.50)] - NSE[DCF(0.90)]} \\ + \frac{NSE[TWS(0.75,0.50)] - NSE[TWS(0.75,0.90)]}{NSE[DCF(0.50)] - NSE[DCF(0.90)]} \\ + \frac{NSE[TWS(0.50,0.50)] - NSE[TWS(0.50,0.90)]}{NSE[DCF(0.50)] - NSE[DCF(0.90)]} \end{array} \right\}/3 \qquad (4)$$

where the numerators, expressed as $NSE[\cdot, \cdot] - NSE[\cdot, \cdot]$, contain the gradients in the hydrological prediction skill between IHC (DCF) weight values of 0.5, 0.75, and 0.9.

## Code availability

Statistical methods are illustrated through text and figure captions. The analyzing data and drawing plots computer codes are in Fortran or NCAR Command Language (NCL) scripts. All the scripts are available upon request.

## Data availability

The CRUNCEP forcing data are available at UCAR website (https://svn-ccsm-inputdata.cgd.ucar.edu/trunk/inputdata/atm/datm7/). The other four precipitation datasets (CPC, GPCC, GPCP, and PREC) are available at (https://www.esrl.noaa.gov/psd/data/gridded/). The CMIP5 decadal hindcast data was provided by the World Climate Research Program's Working Group on Coupled Modeling (http://cmip-pcmdi.llnl.gov/cmip5/availability.html). The CLM4.5 is available at CESM website (http://www.cesm.ucar.edu/models/cesm1.2/).

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

## Acknowledgements

This work was supported by National Key R&D Program of China (2018YFA0606002), National Natural Science Foundation of China (41875105), the Startup Foundation for Introducing Talent of NUIST, and CAS Key Research Program of Frontier Sciences (QYZDY-SSW-DQC012).

## Author contributions

X.Y. conceived and designed the study. E.Z. conducted the simulations and performed the analyses. E.Z. and X.Y. wrote the paper. E.Z. and X.Y. are co-first authors. A.W. provided critical insights on the results interpretation.

## Additional information

**Competing interests:** The authors declare no competing interests.

