## [Peer Review File · Nature Communications]

Reviewers' comments:

Reviewer #1 (Remarks to the Author):

When I first read the title and abstract I was initially a little surprised that this topic could be the focus of a Nature Communications paper, because it seemed rather specialised (e.g. the focus on 'elasticity'). However, after reading the paper closely I am impressed and think the work is important, of high quality, and certainly of general interest to anyone in water-related fields.

The authors explain how predictability of terrestrial water storage (TWS) is currently restricted over decadal timescales, because it is limited by our knowledge of hydrological predictability and forecast skill over these same time periods. The paper claims that 'the era for decadal hydrological prediction is expected in the near future' and that this paper will provide the benchmark skill which is now required to guide and improve future decadal hydrological predictions. I agree with these claims and consider the work will be of interest to those in hydrological forecasting and the paper will indeed constitute a benchmark for future analyses, if the authors can provide some additional information (suggestions are provided below).

The statistical analyses are certainly appropriate (based on well-established methods) and valid, and the paper is well written and clear to follow. I believe a little more detail could be included to allow the work to be fully reproducible (suggestions below).

*Major points

The concept of 'forecast skill elasticity' (abstract & l.76) is not well explained at the start and therefore the paper is not immediately clear to non-specialists. The definition that is given at l.123-125 ('a larger elasticity ... means...') could come earlier in the paper.

Using only the NSE to represent the skill is potentially a little misleading. Even if the correlation coefficients are low, I think they should be provided to the readers, for the full range of timescales (not just 4-year averages), in the supplementary materials. This would provide a real benchmark.

In the methods, it is unclear to me why 4-year averages are used. This needs to be clearly justified.

The paper focusses mainly on physical-dynamical forecasting using a land surface model (CLM4.5) and decadal hindcasts of 10 CMIP5 models. However, the use of a land surface model is only mentioned at the end of the paper in the last paragraph (l.205). I think it needs to be explicit from the start of the paper what type of approach is used and how it differs from other approaches (e.g. Mendoza et al. 2017) including new statistical-dynamical approaches (e.g. Slater and Villarini 2018) which may also be a major avenue for decadal predictions in the future. This will help make the paper more accessible to non-specialists.

The colour schemes in the figures probably need to be revised because they are indistinguishable for someone with colour-blindness or when viewed/printed in greyscale. Also, given that the scales range from low to high it seems like the colours could be shades of a similar colour, rather than passing through the entire spectrum of colours?

*Minor points

L.15. The abstract does not mention the use of CMIP5 models or the use of a land surface model. Perhaps this is worth mentioning, to make the paper more accessible.

L.47. 'semiarid regions where the hydrological variability is not negligible' could be more direct/explicit

L.53. 'they' = ?
L.79. 'representing the gradients': this is a bit vague
L.88. 'uncoupled climate-hydrology approach': also a bit vague
L.99. 'together with a' : remove 'a'
L.101/275. prefect -> perfect
L.102. "climo"?
L.113-120. This reads like you are suggesting that latitude is the primary factor in determining TWS skill sensitivity to increasing IHC skill. Please explain/clarify.
L.291. 'for each 10-years simulation' -> for each 10-year simulation
L.303. 'Because of random resampling, we did a 9-point smoothing..': please clarify.
L.450. 'averaged over three selected basins': do you mean the basin-averaged values?
L.458/462: '4 years mean' -> '4-year mean'
L.459: 'a value of 0.8 represents': -> 'For example, a value of 0.8 means that a 100% improvement..'

***Figures**

Figure 1 & others. The rainbow colour is pretty but likely not discernible for people with reading difficulties or when printed in greyscale.

Figure 3. 'The bias was removed': please clarify; I can't see the explanation or description of bias removal.

Figure 4. If possible, it might be worth making the maps a little larger (e.g. by reducing the white space between rows) so we can better distinguish the basins.

***Supporting materials**

The figures provided in the supporting materials are very helpful.

***References**

Mendoza, P.A., Wood, A.W., Clark, E., Rothwell, E., Clark, M.P., Nijssen, B., Brekke, L.D. and Arnold, J.R., 2017. An intercomparison of approaches for improving operational seasonal streamflow forecasts. *Hydrology and Earth System Sciences*, 21(7), p.3915. <https://www.hydrology-earth-syst-sci.net/21/3915/2017/hess-21-3915-2017.pdf>

Slater, L.J. and Villarini, G., 2018. Enhancing the predictability of seasonal streamflow with a statistical-dynamical approach. *Geophysical Research Letters*.
<https://agupubs.onlinelibrary.wiley.com/doi/abs/10.1029/2018GL077945>

Reviewer #2 (Remarks to the Author):

Please see the attached review report.

Reviewer #2 (Remarks to the Author) attached review report

September 17, 2018

Review for Nature Communications

Title: Benchmark decadal forecast skill for terrestrial water storage estimated by an elasticity framework

Authors: Zhu, Yuan, Wood

This paper examines how the forecast skill for terrestrial water storage (TWS) depends on the accuracy of initial hydrological conditions (IHCs) and decadal climate forecasts (DCF). The authors demonstrate that IHCs are more important than DCFs for short-lead forecasts, whereas DCFs are more important for longer leads. The results are potentially important for decadal TWS prediction. However, some of the analyses and discussions need to be improved. In particular, the authors may want to address the usefulness of decadal precipitation forecasts from CMIP5 models. Therefore, I suggest that the manuscript could be published pending major revisions. Specific comments are listed below.

Major comments:

1. The skill of decadal forecast with current climate models is relatively low, especially for precipitation, which is at a preliminary and experimental stage. I really doubt that beyond seasonal timescale, longer-lead prediction forecast has any predictive value for TWS. The authors may want to verify the precipitation forecasts by showing the time series of decadal precipitation forecasts along with the time series of observed precipitation for each river basin.
2. Line 91, first paragraph of “Results”: Please explain explicitly what IHCs and DCFs (variables) are used in the ESP and rev-ESP.
3. Line 104, forecast lead time: The forecast lead time used in the manuscript may be different from that commonly used in climate predictions. For example, for 10-year continuous forecasts, the average of year 1 to year 4 forecasts is defined as 0-year lead forecast, year 2 to year 5 as 1-year lead forecast, and year 7 to year 10 as 6-year lead forecast.
4. Lines 156-157: The authors claim that “the prediction skill does not necessarily decrease over leads, which might be caused by the perturbation from external forcings.” It should be noted that these external forcings, such as volcanic eruptions, are unpredictable. They do not help increase the real-time forecast skill prior to their occurrence (e.g., Mehta et al. 2018: Climate Dynamics, <https://doi.org/10.1007/s00382-018-4321-1>).
5. Line 229: It is not clear about how the CMIP5 model precipitation is used in this study. Is it multi-model ensemble mean? Are all models equally weighted? How many ensemble members are used from individual models? Without detailed descriptions, it is hard to reproduce the work.
6. Line 233-238, CRUNCEP: The authors should use observation data “to assess the performance of precipitation predictions from CMIP5 models”, instead of CRUNCEP. There are several observational precipitation datasets available (e.g., <https://www.esrl.noaa.gov/psd/data/gridded/>). “CRUNCEP” should be “CRUNCEP”.

7. Lines 242-246: The descriptions are confusing (i^{th} starting year, lead q years, total number of starting years).

8. Lines 459-460: How are the “100% improvement in the accuracy of DCFs or IHCs” and “80% increase in TWS forecast skill” defined?

9. Line 464: “The bias for CMIP5 model-prediction was removed before calculating NSE”. Is it the mean bias of the models? Does the bias depend on lead time? Are the variables in the right side of Eq. 1 bias-corrected?

Minor comments/edits:

1. Line 46: a third **of** the world’s land areas

2. Line 82: Insert “an” between “such” and “elasticity” (such **an** elasticity framework).

3. Line 241: Are all the variables annual means?

Responses to the comments from Reviewer #1

We thank the reviewer for the critical review. The thoughtful comments have helped improve the manuscript. The reviewer's comments are italicized and our responses immediately follow.

When I first read the title and abstract I was initially a little surprised that this topic could be the focus of a Nature Communications paper, because it seemed rather specialised (e.g. the focus on 'elasticity'). However, after reading the paper closely I am impressed and think the work is important, of high quality, and certainly of general interest to anyone in water-related fields.

The authors explain how predictability of terrestrial water storage (TWS) is currently restricted over decadal timescales, because it is limited by our knowledge of hydrological predictability and forecast skill over these same time periods. The paper claims that 'the era for decadal hydrological prediction is expected in the near future' and that this paper will provide the benchmark skill which is now required to guide and improve future decadal hydrological predictions. I agree with these claims and consider the work will be of interest to those in hydrological forecasting and the paper will indeed constitute a benchmark for future analyses, if the authors can provide some additional information (suggestions are provided below).

The statistical analyses are certainly appropriate (based on well-established methods) and valid, and the paper is well written and clear to follow. I believe a little more detail could be included to allow the work to be fully reproducible (suggestions below).

Response: We would like to thank the reviewer for the positive comments. Please see our responses below.

The concept of 'forecast skill elasticity' (abstract & l.76) is not well explained at the start and therefore the paper is not immediately clear to non-specialists. The definition that is given at l.123-125 ('a larger elasticity ... means...') could come earlier in the paper.

Response: Thanks for the comments. We have revised it accordingly as follows:

“The forecast skill elasticities²² (i.e., Eqs. 3 and 4 in Methods) represent the gradients in hydrological forecast skill (e.g., streamflow forecast skill) relative to the accuracy for predictors (e.g., IHCs or meteorological forcings), and a larger elasticity of a predictability source means a larger contribution to the improvement of the hydrological prediction given an improved skill in the predictability source.” (L86-92 in the tracked version of the revised manuscript)

Using only the NSE to represent the skill is potentially a little misleading. Even if the correlation coefficients are low, I think they should be provided to the readers, for the full range of

timescales (not just 4-year averages), in the supplementary materials. This would provide a real benchmark.

In the methods, it is unclear to me why 4-year averages are used. This needs to be clearly justified.

Response: Thanks for the comments. We have now reproduced Figs. 1-4 by using correlation coefficient (CC) to represent the skill of IHCs, DCFs and TWS, and provided the results in the supplementary material (Figs. S4, S6-S8). The results show that conclusions based on the two metrics are similar, with marginal differences over specific river basins. We have revised the manuscript as follows:

“... The results for the Yangtze basin, a semi-humid river basin, fall between the humid and semiarid basins (Fig. 1). Similar characteristics for skill variations are also found by using correlation as a measure (Supplementary Fig. S4).” (L143-146)

“... the DCFs NSE (correlation) elasticities increase from 0.48 (0.36) to 0.81 (0.73) as lead time increases from 1-4 years to 7-10 years, while the IHCs NSE (correlation) elasticities decrease from 0.55 (0.36) to 0.19 (0.13) (Fig. 2 and Supplementary Fig. S6). DCFs elasticities are larger than one over 15% (22% for correlation) of the basins at 7-10 years lead ...” (L164-168)

“For the 1-4 years lead, there are skillful predictions (NSE>0 and correlation larger than 0.5) over the Amazon basin in South America, the Shatt el Arab basin in Middle East, the Ob and Syr-Darya basins in central Asia, and the Lena basin in Far East (Fig. 3a and Supplementary Fig. S7a).” (L179-184)

“As forecast lead time increases, the benchmark skill averaged over the 32 major river basins decreases from 0.51 (0.67 for correlation) at 1-4 years lead (left columns in Fig. 4a and Supplementary Fig. S8a) to 0.25 (0.42), 0.19 (0.30) and 0.11 (0.17) at leads of 3-6 years, 5-8 years and 7-10 years respectively (left columns in Figs. 4b-4d and Supplementary Figs. S8b-S8d). Compared with ESP forecast skill which has been used for the benchmark in most hydrological applications (right columns in Fig. 4 and Supplementary Fig. 8), the new benchmark skill is significantly ($p<0.05$) higher over 25% (43% for correlation) river basins after incorporating CMIP5 decadal prediction information at 1-4 years lead. The increases are more obvious at longer lead times, with significant NSE (correlation) increases over 31% (56%), 44% (66%) and 47% (59%) basins at leads of 3-6, 5-8, 7-10 years.” (L197-208)

The 4-year average (and similar smoothing periods from 2-10 years) is a common choice in studies analyzing decadal climate prediction, as it filters high-frequency (annual to ENSO-scale) variability but retains the information of climate predictability at a longer time-scales^{1,2,3}. Therefore, we would like to keep using the 4-year averages for the analysis in the article. The

correlation coefficient (CC) benchmark skill for 1-year averages is also provided in this response letter (Figure R1).

Figure S4. The same as Figure 1, but using correlation coefficient (CC) to measure predictive skill of IHCs, DCFs and TWS.

Figure S6. The same as Figure 2, but for DCFs and IHCs elasticities measured by correlation.

a) Lead 1-4 years

b) Lead 3-6 years

c) Lead 5-8 years

d) Lead 7-10 years

Figure S7. The same as Figure 3, but for CMIP5 multi-model ensemble hindcast skill of precipitation measured by correlation. The basins with significant correlations ($p < 0.1$) are dotted.

Figure S8. The same as Figure 4, but for benchmark skill and ESP skill measured by correlation. The regions with benchmark skill significantly ($p < 0.05$) higher than ESP skill are dotted.

Figure R1. The same as Figure S8, but for 1-year mean TWS benchmark decadal hindcast skill (left) and ESP hindcast skill (right) over global major river basins.

The paper focuses mainly on physical-dynamical forecasting using a land surface model (CLM4.5) and decadal hindcasts of 10 CMIP5 models. However, the use of a land surface model

is only mentioned at the end of the paper in the last paragraph (l.205). I think it needs to be explicit from the start of the paper what type of approach is used and how it differs from other approaches (e.g. Mendoza et al. 2017) including new statistical-dynamical approaches (e.g. Slater and Villarini 2018) which may also be a major avenue for decadal predictions in the future. This will help make the paper more accessible to non-specialists.

Response: Thanks for your suggestion. We have revised the manuscript as follows:

“In addition, there is an opportunity to obtain a benchmark forecast skill by incorporating the-state-of-the-art information of DCFs from the fifth Coupled Model Intercomparison Project (CMIP5) decadal hindcasts²⁴. Prior studies and operational applications have employed post-ESP or statistical-dynamical (hybrid and hierarchical) approaches to incorporate climate information into the seasonal hydrological forecasting^{25,26}. Here, we propose a method for estimating a benchmark decadal hydrological prediction skill, which is based on a climate-hydrology approach¹¹ (e.g., physical hydrological model predictions driven by CMIP5 decadal climate predictions), and further use the elasticity framework to quantify predictability gradients. The benchmark skill can provide a new norm or starting point to assess whether different strategies for future upgrades in the climate-hydrology approach¹¹ are beneficial for decadal hydrological prediction.” (L95-107)

The colour schemes in the figures probably need to be revised because they are indistinguishable for someone with colour-blindness or when viewed/printed in greyscale. Also, given that the scales range from low to high it seems like the colours could be shades of a similar colour, rather than passing through the entire spectrum of colours?

Response: Thanks for the thoughtful suggestion. We have changed color schemes of all figures (Figs. 1-4 and Figs. S4, S6-8) to insure they are friendly to more readers.

*Minor points

L.15. The abstract does not mention the use of CMIP5 models or the use of a land surface model. Perhaps this is worth mentioning, to make the paper more accessible.

Response: Thanks for the suggestion. We have revised it as follows:

“Here we use decadal climate hindcasts from Coupled Model Intercomparison Project phase 5 (CMIP5) models and perform hydrological simulations by using Community Land Model (CLM4.5), and to estimate a benchmark decadal forecast skill ...” (L22-24)

L.47. *‘semiarid regions where the hydrological variability is not negligible ’ could be more direct/explicit*

Response: We have revised as “... such as northern China, southern Africa and the Middle East, ...” (L54-55)

L.53. *‘they ’ = ?*

Response: We have clarified as “These internal and external factors affect temperature and precipitation ...” (L61-62)

L.79. *‘representing the gradients ’ : this is a bit vague*

Response: We have revised it as follows:

“The forecast skill elasticities¹¹ (i.e., Eqs. 3 and 4 in Methods) represent the gradients in hydrological forecast skill (e.g., streamflow forecast skill) relative to gradients in accuracy for predictors (e.g., IHCs or meteorological forcings), hence a larger elasticity of a predictability source means a larger contribution to the improvement of the hydrological prediction given an improved skill in the predictability source.” (L86-92)

L.88. *‘uncoupled climate-hydrology approach ’ : also a bit vague*

Response: We meant the standalone hydrological prediction driven by climate prediction, while there is no feedback from land to atmosphere. We have now removed “uncoupled” to avoid confusion.

L.99. *‘together with a ’ : remove ‘a ’*

L.101/275. *prefect -> perfect*

Response: Revised as suggested. (L122, L123, L326)

L.102. *“climo ” ?*

Response: We have clarified as “(‘climo’, abbreviation for ‘climatology’)” (L125)

L.113-120. *This reads like you are suggesting that latitude is the primary factor in determining TWS skill sensitivity to increasing IHC skill. Please explain/clarify.*

Response: Thanks for the comment. The skill sensitivity is related to the climate conditions. We have now replaced geographic locations with climate conditions as follows:

“... the Amazon basin in the humid area ... the Yenisei basin in the semiarid area ... The results for the Yangtze basin, a semi-humid river basin, fall between the humid and semiarid basins (Fig. 1).” (L137; L139; L143-145)

L.291. *‘for each 10-years simulation ’ -> for each 10-year simulation*

Response: Revised as suggested. (L343)

L.303. *‘Because of random resampling, we did a 9-point smoothing.. ’ : please clarify.*

Response: We have clarified as follows:

“Because the random resampling produced random biases, we did a 9-point smoothing along the y-axis to generate a more uniform response surface.” (L354-356)

L.450. *‘averaged over three selected basins ’ : do you mean the basin-averaged values?*

Response: Yes, and we have revised as “4-year mean basin-averaged terrestrial water storage (TWS) over three selected basins ...” (L530-531)

L.458/462: *‘4 years mean ’ -> ‘4-year mean ’* (L530, L540, L547)

L.459: *‘a value of 0.8 represents ’ : -> ‘For example, a value of 0.8 means that a 100% improvement.. ’*

Response: Revised as suggested. (L541-543)

**Figures*

Figure 1 & others. The rainbow colour is pretty but likely not discernible for people with reading difficulties or when printed in greyscale.

Response: We have changed the color schemes to assure they are discernible to more readers.

Figure 3. ‘The bias was removed ’ : please clarify; I can ’t see the explanation or description of bias removal

Response: We have clarified as follows:

“... (we use anomaly time series to avoid the impact of predictions bias),...” (L294-295)

“The anomaly of CMIP5 model-predicted precipitation to CMIP5 climatology was used to calculating NSE, for circumvent the bias in the interpretation of results.” (L548-550)

Figure 4. If possible, it might be worth making the maps a little larger (e.g. by reducing the white space between rows) so we can better distinguish the basins.

Response: Revised as suggested.

**Supporting materials*

The figures provided in the supporting materials are very helpful.

Response: Thanks for your positive comments.

Responses to the comments from Reviewer #2

We thank the reviewer for the critical review. The thoughtful comments are valuable and the suggestions make our work more confident. The reviewer's comments are italicized and our responses immediately follow.

This paper examines how the forecast skill for terrestrial water storage (TWS) depends on the accuracy of initial hydrological conditions (IHCs) and decadal climate forecasts (DCFs). The authors demonstrate that IHCs are more important than DCFs for short-lead forecasts, whereas DCFs are more important for longer leads. The results are potentially important for decadal TWS prediction. However, some of the analyses and discussions need to be improved. In particular, the authors may want to address the usefulness of decadal precipitation forecasts from CMIP5 models. Therefore, I suggest that the manuscript could be published pending major revisions. Specific comments are listed below.

Response: We would like to thank the reviewer for the positive comments and valuable suggestion. Please see our responses below.

Major comments:

1. The skill of decadal forecast with current climate models is relatively low, especially for precipitation, which is at a preliminary and experimental stage. I really doubt that beyond seasonal timescale, longer-lead prediction forecast has any predictive value for TWS. The authors may want to verify the precipitation forecasts by showing the time series of decadal precipitation forecasts along with the time series of observed precipitation for each river basin.

Response: We agree with the reviewer that the decadal forecast skill of precipitation with climate models is relatively low, which are shown in the time series of observed and predicted precipitation in Figure S10. However, skillful precipitation predictions have been found for specific basins (e.g., Lena, Yenisei) at given lead times, based on the metrics of both NSE (Figure 3) and correlation (Figure S7). We have used the precipitation prediction information to help estimate a benchmark decadal forecast skill for TWS as a useful reference point in the elasticity analysis framework. Nevertheless, how to incorporate these kinds of useful climate prediction information into the “real” hydrological prediction is an open question, where our benchmark skill provides a reference for the assessment of the hydrological prediction.

Figure S10. Time series of CMIP5 multi-model ensemble 4-year mean precipitation hindcasts and CRUNCEP 4-year mean precipitation observations averaged over 32 basins. The brown lines and red-blue lines stand for CRUNCEP and CMIP5, respectively.

Figure 3. Decadal hindcast skill (NSE) for CMIP5 multi-model predicted 4-year mean precipitation averaged over global major river basins at different lead times. The anomaly of CMIP5 model-predicted precipitation to CMIP5 climatology was used to calculating NSE, for bias correction.

a) Lead 1-4 years

b) Lead 3-6 years

c) Lead 5-8 years

d) Lead 7-10 years

Figure S7. The same as Figure 3, but for CMIP5 multi-model ensemble hindcast skill of precipitation measured by correlation. The basins with significant correlations ($p < 0.1$) are dotted.

2. Line 91, first paragraph of “Results”: Please explain explicitly what IHCs and DCFs (variables) are used in the ESP and rev-ESP.

Response: We have now explained as follows:

“The ESP and rev-ESP experiments were carried out by performing CLM4.5 model simulations. In these experiments, the DCFs include atmospheric temperature, humidity, wind and pressure near the surface, radiation, and precipitation, while IHCs represent the initial states of TWS, soil moisture and soil temperature, etc.” (L112-116 in the tracked version of the revised manuscript)

3. Line 104, forecast lead time: The forecast lead time used in the manuscript may be different from that commonly used in climate predictions. For example, for 10-year continuous forecasts, the average of year 1 to year 4 forecasts is defined as 0-year lead forecast, year 2 to year 5 as 1-year lead forecast, and year 7 to year 10 as 6-year lead forecast.

Response: These forecast lead times (e.g., 1-4 years, 2-5 years) are widely used in the decadal climate prediction community^{1,2,3}. However, we agree with the reviewer that they are different from those used in weather and seasonal forecasting. Therefore, we have now clarified them as follows:

“(here, “1-4 years lead prediction” represents 4-year average prediction with 0-year lead, “2-5 years lead prediction” represents that with 1-year lead, and so on)” (L127-129)

4. Lines 156-157: The authors claim that “the prediction skill does not necessarily decrease over leads, which might be caused by the perturbation from external forcings.” It should be noted that these external forcings, such as volcanic eruptions, are unpredictable. They do not help increase the real-time forecast skill prior to their occurrence (e.g., Mehta et al. 2018: *Climate Dynamics*, <https://doi.org/10.1007/s00382-018-4321-1>).

Response: Thanks for the comment. We have revised the statement as “The prediction skill does not necessarily decrease over leads, which might be caused by perturbations from external forcings²⁴. However, these external forcings, such as volcanic eruptions, are unpredictable before their occurrence²⁵, and through rare, their effects on the skill variation may be amplified given limited decadal hindcast samples.” (L186-190)

5. Line 229: It is not clear about how the CMIP5 model precipitation is used in this study. Is it multi-model ensemble mean? Are all models equally weighted? How many ensemble members are used from individual models? Without detailed descriptions, it is hard to reproduce the work.

Response: Thanks for your questions. We have now clarified as follows:

“Precipitation outputs from the decadal hindcasts of 10 CMIP5 models, i.e., bcc-csm 1-1, MRI-CGCM3, CNRM-CM5, EC-EARTH, IPSL-CM5A, FGOALS-g2, FGOALS-s2, MIROC4h, MPI-ESM-LR, MPI-ESM-MR, were equally averaged to form a multi-model ensemble mean to estimate decadal prediction skill for precipitation³³, which was used as the DCFs skill in this work. For each CMIP5 model, the r1i1p1 realization was used.” (L266-271)

6. Line 233-238, *CRUNECP*: The authors should use observation data “to assess the performance of precipitation predictions from CMIP5 models”, instead of *CRUNCEP*. There are several observational precipitation datasets available (e.g., <https://www.esrl.noaa.gov/psd/data/gridded/>). “*CRUNECP*” should be “*CRUNCEP*”.

Response: Thanks for the comment. We used *CRUNCEP* data because it provides all forcing variables (including precipitation, temperature, wind etc.) for the long-term land surface model simulations, as well as the ESP and rev-ESP experiments. To be consistent, we used *CRUNCEP* precipitation in this study as a reference for validating CMIP5 precipitation and estimating the benchmark skill. According to the reviewer’s suggestion, we have now compared it with other 4 precipitation products, and revised the manuscript as follows:

“The dataset is a widely used gauge-based product, and its monthly precipitation values are the same as the CRU data. As compared with other precipitation products based on gauge-observation or satellite-estimates, including CPC, GPCC, GPCP and PREC, regional differences do exist (Supplementary Fig. S9). However, to be consistent with ESP and rev-ESP experiments for elasticity and benchmark skill estimation which are based on *CRUNCEP*-driven CLM4.5 simulations, we kept using *CRUNCEP* as a reference for validating CMIP5 decadal prediction of precipitation.” (L276-283)

Figure S9. Zonal mean precipitation from five datasets. Rain-gauges and satellite-based algorithms are used to produce Climate Prediction Center Merged Analysis of Precipitation (CPC) and Global Precipitation Climatology Centre (GPCP) monthly precipitation analysis product, while Precipitation Reconstruction over Land (PREC), Global Precipitation Climatology Centre (GPCC) and CRUNCEP datasets are based on a number of station observation.

7. Lines 242-246: The descriptions are confusing (*i*th starting year, lead *q* years, total number of starting years).

Response: We have clarified as “... where *q* is forecast lead (years; e.g., 1-4 years, 2-5 years), *i* stands for the *i*th initialization year (i.e., 1961, 1966, 1971, 1976, 1981, 1986, 1991, 1996, 2001, 2006) ...and *n* (=10 in this study) is the total number of initialization years...” (L290-296)

8. Lines 459-460: How are the “100% improvement in the accuracy of DCFs or IHCs” and “80% increase in TWS forecast skill” defined?

Response: We have clarified as “A value of 0.8 represents that 100% improvement in the accuracy of DCFs or IHCs (e.g., NSE increases from 0.2 to 0.4) would result in 80% increase in TWS forecast skill (e.g., NSE increases from 0.3 to 0.54).” (L541-543)

9. Line 464: “The bias for CMIP5 model-prediction was removed before calculating NSE”. Is it the mean bias of the models? Does the bias depend on lead time? Are the variables in the right side of Eq. 1 bias-corrected?

Response: Yes, we translate the timeseries into anomalies by removing the mean, which circumvents the inclusion of bias in the calculation of NSE. The bias varies with both lead time and location. The variables in the right side of Eq. 1 are bias-corrected, where anomalies of the original variables are used for the NSE calculation. We have revised the Eq. 1 and its clarification as follows:

$$“NSE_q = 1 - \frac{\sum_{i=1}^n (X_{i,q}^{obs} - X_{i,q}^{sim})^2}{\sum_{i=1}^n (X_{i,q}^{obs})^2}, \quad (1)”$$

where q is forecast lead (years; e.g., 1-4 years, 2-5 years), i stands for the i^{th} initialization year (i.e., 1961, 1966, 1971, 1976, 1981, 1986, 1991, 1996, 2001, 2006), so $X_{i,q}^{obs}$ is the anomaly of observation for the i^{th} initialization year at lead q years for the concerned component (e.g., precipitation or TWS), $X_{i,q}^{sim}$ is the same as $X_{i,q}^{obs}$ but for the anomaly of simulations or predictions (we use anomaly to remove bias of predictions), and n (=10 in this study) is the total number of initialization years in the CMIP5 decadal hindcasts.” (L289-297)

Minor comments/edits:

1. Line 46: a third *of* the world’s land areas

2. Line 82: Insert “an” between “such” and “elasticity” (such an elasticity framework).

Response: Revised as suggested. (L53; L93)

3. Line 241: Are all the variables annual means?

Response: They can annual means or 4-year averages depending on the assessment purposes.

References:

1. Mohino, E., Keenlyside, N. & Pohlmann, H., Decadal prediction of Sahel rainfall: where does the skill (or lack thereof) come from? *Clim. Dyn.* **47**, 3593-3612 (2016).
2. Bellucci, A. *et al.*, An assessment of a multi-model ensemble of decadal climate predictions. *Clim. Dyn.* **44**, 2787-2806 (2015).
3. Mehta, V.M. *et.al.*, Predictability of phases and magnitudes of natural decadal climate variability phenomena in CMIP5 experiments with the UKMO HadCM3, GFDL-CM2.1, NCAR-CCSM4, and MIROC5 global earth system models. *Clim. Dyn.* <https://doi.org/10.1007/s00382-018-4321-1> (2018)
4. Mendoza, P.A., Wood, A.W. *et.al.*, An intercomparison of approaches for improving operational seasonal streamflow forecast. *Hydro. Earth Syst. Sci.*, **21**, 3915-3935 (2017)
5. Slater, L.J. & Villarini, G. Enhancing the predictability of seasonal streamflow with a statistical-dynamical approach. *Geophys. Res. Lett.*, **45**, 6504-6513 (2018)

REVIEWERS' COMMENTS:

Reviewer #1 (Remarks to the Author):

I thank the authors for their detailed and thorough response to queries.

I find the authors have considerably improved and clarified many parts of their manuscript. In doing so they have made the manuscript much more accessible, especially for readers outside the immediate field.

I find this is a valuable and very interesting contribution to the literature and recommend publication.

Noticed a few very minor typos:

L.24 "and to estimate" – check the sentence structure – something doesn't make sense

L.142. SupplEmentary

Figure 3. to calculate NSE, to circumvent the bias

Reviewer #2 (Remarks to the Author):

The authors have taken considerable steps in revising the manuscript, and comprehensively addressing my earlier comments. I recommend acceptance of the paper.

Figure maps werecreated by using the NCAR
21 Command Language (Version 6.3.0) [Software]. (2016). Boulder, Colorado:
22 UCAR/NCAR/CISL/TDD. <http://dx.doi.org/10.5065/D6WD3XH5>.